# Learned learning rate schedules for deep neural network training using reinforcement learning

**Shreyas Subramanian, Vignesh Ganapathiraman, & Aly El Gamal** [*]
Amazon.com, Inc., 410 Terry Ave N, Seattle 98109, WA
{subshrey,vignesga,alyeg}@amazon.com

## Abstract

We present a novel strategy to generate learned learning rate schedules for any optimizer using reinforcement learning (RL). Our approach trains a Proximal Policy Optimization (PPO) agent to predict optimal learning rate schedules for SGD, which we compare with other optimizer-scheduler combinations and full grid search. Our experiments show that the agent learns to generate dynamic schedules that result in stable, non-divergent loss histories, and can be more useful in practice than equally-expensive Hyperparameter Optimization and fixed optimizer-scheduler combinations.

## 1 Introduction

The optimization of Deep Neural Networks (DNNs) has been a long-standing challenge in the field of machine learning. One of the critical hyperparameters is the learning rate schedule, which determines the step size for each iteration of the optimization process. A common approach for setting the learning rate schedule is to use a fixed schedule, such as a step decay or a cosine decay (Loshchilov & Hutter, 2016). However, these fixed schedules may not be optimal for all DNNs and tasks, and they may require careful tuning to achieve good performance (Ruder, 2016).

Recently, there has been growing interest in learned optimization algorithms, which rely on a meta-optimization process to learn an optimization algorithm from data (Wichrowska et al., 2017; Metz et al., 2022). These algorithms have been shown to improve the stability and generalization performance of DNNs compared to traditional optimization algorithms (Andrychowicz et al., 2016; Metz et al., 2019). Additionally, learned optimization algorithms have the ability to adapt to different tasks and architectures, making them a versatile tool for optimizing DNNs (Andrychowicz et al., 2016). In recent years, several works have demonstrated the effectiveness of learned optimization algorithms for training DNNs (Wichrowska et al., 2017; Metz et al., 2019). Reinforcement learning (RL) has also been applied to learned optimization, with the goal of learning an optimal policy for adjusting the optimization parameters during training (Bello et al., 2017; Metz et al., 2021; 2020). In this paper, we develop a novel learned scheduler for SGD using RL. Our approach trains a Proximal Policy Optimization (PPO) model to predict the optimal learning rate schedules for SGD by using the training loss and other information that encapsulates the state of the training. Our results demonstrate that the schedules generated by the RL agent result in more stable convergence and lower validation loss compared to popular fixed learning rate schedules, and that the learned optimizer can be used as a computationally efficient alternative to full grid search methods. Furthermore, our approach is generally applicable to any optimizer.

## 2 Learned schedules using RL

In this work, we develop a novel learned schedule for SGD. To this end, we train a RL agent to predict the optimal learning rate of SGD at each iteration (the learning schedule). The agent observes the following signals 1) *validation loss from previous epoch*, 2) *distance to the end of training*

---

[*]

$(\mathrm{epoch}_{\max} - \mathrm{epoch})$ and 3) layer-wise gradient norms of the deep learning model. We device the following simple reward function to encourage faster convergence and penalize loss divergence.

$$r = \gamma(\mathrm{best\_loss} - \mathrm{val\_loss}) + \lambda(\mathrm{epoch}_{\max} - \mathrm{epoch}), \tag{1}$$

where $\lambda, \gamma$ are hyperparameters and $\mathrm{best\_loss}$ is the lowest validation loss achieved so far. We also penalize the agent when divergence is detected (NaN's in weights or loss).

## 3   EXPERIMENTS AND RESULTS

Via deep learning [1] for the MNIST and CIFAR-100 dataset, we compare the results obtained by the RL-learned schedule with SGD to that of SGD, Adadelta and Adam with a *ConstantLR* and *OneCycleLR* (Smith & Topin, 2019) schedulers using grid search. Optimzers and schedulers used are implementations that exist in PyTorch [2]. Note that these experiments, even for small datasets are expensive to run - we do a full grid search at 100 initial LRs for 100 epochs, repeat the experiment with 4 different starting LRs ($5e^{-4}, 1e^{-3}, 5e^{-3}, 1e^{-2}$) for four different optimizers, and two base schedulers noted above. When comparing grid search, optimizer-schedule, and agent training we repeat trained agent runs 5 times with different seeds. We use the same computational budget (total optimizer steps $= 10000$) for grid search vs. RL, and report the best and average loss along with divergence statistics. Separately, we also tested the learned agent's performance with a full line search method (SLS from (Vaswani et al., 2019), see Appendix).

Across experiments, we observe that 1) the agent learns a fixed policy that produces dynamic schedules that work for multiple initial seeds 2) the agent schedule results in more stable runs resulting in the best possible final loss compared to other optimizer-scheduler combinations that may either diverge, or result in a high final loss; and 3) in cases where the initial LR is too high, no scheduler appears to converge including the trained agent, which again only provides a more stable schedule.

We include results from two runs below in Table 1 for MNIST and CIFAR-100 datasets. Once again we note that the agent is rewarded for finding non-divergent, stable, and good performing schedules which is more valuable in practice than sensitive optimizer-scheduler combinations that may diverge (see below for higher loss value when a scheduler is added.)

| Dataset | Agent | SGD | Adadelta | Adam | SGD-Sch | Adadelta-Sch | Adam-Sch |
|---------|-------|-----|----------|------|---------|--------------|----------|
| MNIST | 0.038 | 0.035 | 0.025 | 0.013 | 0.081 | 0.57 | 0.030 |
| CIFAR 100 | 0.033 | 0.026 | 0.026 | 0.030 | 0.39 | 0.044 | 2.30 |

Table 1: Final Loss for optimizers with the MNIST and CIFAR100 datasets; "Sch" is OneCycleLR

**Additional observations.**   Further, we observe (see Figure 2 in the Appendix) that some optimizer-scheduler combinations can lead to divergence or sub-optimal final loss values. This highlights the advantage of our proposed RL strategy as it finds a stable schedule that works across several initial LRs and seeds, without the need for the manual tuning trial and error effort. We also note that upon investigating the RL action history, we find that the agent often finds a non-trivial schedule (see Figures 3 to 5 in Appendix) where dramatic increases and decreases in the learning rate frequently take place, alluding to the complexity of the alternative manual tuning task. Lastly we see that the learned policies for the learning rate schedule are sensitive the RL agent hyperparameters, such as PPO learning rate and number of iterations.

## 4   CONCLUSION

This paper presents a novel method for finding learned LR schedules for any optimizer using reinforcement learning as an alternative to standard Hyperparameter Optimization methods. Experiments on image classification tasks showed that the schedules generated by the RL agent resulted in stable convergence and lower validation loss compared to popular fixed learning rate schedules.

---

[1]Implementation from `https://github.com/pytorch/examples/blob/main/mnist`
[2]Pytorch optimizer class `https://pytorch.org/docs/stable/optim.html`

## URM STATEMENT

The authors acknowledge that at least one key author of this work meets the URM criteria of ICLR 2023 Tiny Papers Track.

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

## A APPENDIX

### A.1 POMDP FORMULATION

We assume that our problem is a Partially Observable Markov Decision Process (POMDP) with the following properties:

1. State $s$ is the set of model network weights, gradients, current loss value, current epoch

2. Observations $o$ are the gradient norms, current loss value, and current epoch value

3. Actions $a$ is the Learning rate $LR$ for the upcoming epoch. This is a global LR, and not a parameter specific, or parameter-group specific LR. We also conducted experiments where the action space included other controllable parameters of the optimizer (such as weight decay or momentum), but we do not include these results here

4. Reward $r$ is a function of current loss value, the best loss so far and the number of remaining epochs as shown in Eq. 3

The Agent at each step $t$ observes $o$, a part of the environment's *state* $s_t \in s$ and it selects an action (Learning rate) $a_t \in a(s)$. Then, as a consequence of its action the agent receives a *reward*, $r_{t+1} \in r \in \mathbb{R}$. Finally the agent learns the following *policy*, which we treat as a the learned learning rate schedule:

$$\pi_t(s|a) \qquad (2)$$

That is the probability of select an action (here, the learning rate) $a_t = a$, if $s_t = s$.

### A.2 ADDITIONAL DEEP LEARNING EXPERIMENTS

Note that for all experiments we penalize the agent with a large random number from 1000 to 2000 (i.e. $1000 + rand \cdot 1000$), and use $\lambda = 1, \gamma = 10$ in Eq. 3. We allow the PPO agent to explore LRs from $1e-6$ to 1.0, and use a fixed batch size for the agent learning. The LR for the agent matches the initial LR chosen for the experiment; we note that this is not necessary, and is considered another hyperparameter to be carefully chosen. Our experiments show that a dynamic policy that generates LR schedules can be more stable and useful for practical deep learning compared to fixed HPOs that are equally-expensive. Given the large computational requirements for each one of these experiments, we continue to work on testing more optimizer-scheduler combinations, more agent architectures and datasets, along with a stronger theoretical foundation for RL policies for schedules as an alternative to equally-expensive Hyperparameter Optimization.

### A.3 CONSTANTLR COMPARISONS

Table 2: Scheduler - constantLR — Agent LR = 5e-4

|   | Method | Best Final Loss | Avg Final Loss | Divergent case exists |
|---|---|---|---|---|
| 0 | Agent | 0.089957 | 0.089957 | no |
| 1 | SGD | 0.034583 | 0.189786 | yes |
| 2 | Adadelta | 0.030426 | 0.158178 | yes |
| 3 | Adam | 0.131676 | 2.348156 | yes |
| 4 | SGD-Sch | 0.060213 | 0.060213 | no |
| 5 | Adadelta-Sch | 0.105236 | 0.105236 | no |
| 6 | Adam-Sch | 0.064445 | 0.064445 | no |

### A.4 ONECYCLELR COMPARISONS

The tables below show detailed results for optimizer-scheduler combinations, including best and average final loss results.

Table 3: Scheduler - constantLR — Agent LR = 1e-3

|   | Method | Best Final Loss | Avg Final Loss | Divergent case exists |
|---|--------|-----------------|----------------|-----------------------|
| 0 | Agent | 0.041140 | 0.041140 | no |
| 1 | SGD | 0.030961 | 0.200702 | yes |
| 2 | Adadelta | 0.029443 | 0.157513 | yes |
| 3 | Adam | 0.137187 | 2.279212 | yes |
| 4 | SGD-Sch | 0.066977 | 0.066977 | no |
| 5 | Adadelta-Sch | 0.107489 | 0.107489 | no |
| 6 | Adam-Sch | 2.301556 | 2.301556 | no |

Table 4: Scheduler - constantLR — Agent LR = 5e-3

|   | Method | Best Final Loss | Avg Final Loss | Divergent case exists |
|---|--------|-----------------|----------------|-----------------------|
| 0 | Agent | 0.041140 | 0.041140 | no |
| 1 | SGD | 0.030961 | 0.200702 | yes |
| 2 | Adadelta | 0.029443 | 0.157513 | yes |
| 3 | Adam | 0.137187 | 2.279212 | yes |
| 4 | SGD-Sch | 0.066977 | 0.066977 | no |
| 5 | Adadelta-Sch | 0.107489 | 0.107489 | no |
| 6 | Adam-Sch | 2.301556 | 2.301556 | no |

Table 5: Scheduler - constantLR — Agent LR = 1e-2

|   | Method | Best Final Loss | Avg Final Loss | Divergent case exists |
|---|--------|-----------------|----------------|-----------------------|
| 0 | Agent | 0.038207 | 0.038207 | no |
| 1 | SGD | 0.032796 | 0.321682 | yes |
| 2 | Adadelta | 0.027578 | 0.155881 | yes |
| 3 | Adam | 0.111969 | 2.371107 | yes |
| 4 | SGD-Sch | 0.067073 | 0.067073 | no |
| 5 | Adadelta-Sch | 0.111244 | 0.111244 | no |
| 6 | Adam-Sch | 2.301887 | 2.301887 | no |

Table 6: Scheduler - OneCycleLR — Agent LR = 5e-4

|   | Method | Best Final Loss | Avg Final Loss | Divergent case exists |
|---|--------|-----------------|----------------|-----------------------|
| 0 | Agent | 0.038413 | 0.038413 | no |
| 1 | SGD | 0.035030 | 0.213103 | yes |
| 2 | Adadelta | 0.025518 | 0.158193 | yes |
| 3 | Adam | 0.133940 | 2.305509 | yes |
| 4 | SGD-Sch | 0.081351 | 0.081351 | no |
| 5 | Adadelta-Sch | 0.566937 | 0.566937 | no |
| 6 | Adam-Sch | 0.030448 | 0.030448 | no |

Table 7: Scheduler - OneCycleLR — Agent LR = 1e-3

|   | Method | Best Final Loss | Avg Final Loss | Divergent case exists |
|---|--------|-----------------|----------------|-----------------------|
| 0 | Agent | 0.033544 | 0.033544 | no |
| 1 | SGD | 0.031457 | 0.339102 | yes |
| 2 | Adadelta | 0.028417 | 0.159335 | yes |
| 3 | Adam | 0.126896 | 2.235273 | yes |
| 4 | SGD-Sch | 0.091012 | 0.091012 | no |
| 5 | Adadelta-Sch | 0.683922 | 0.683922 | no |
| 6 | Adam-Sch | 0.029626 | 0.029626 | no |

Table 8: Scheduler - OneCycleLR — Agent LR = 5e-3

|   | Method | Best Final Loss | Avg Final Loss | Divergent case exists |
|---|--------|-----------------|----------------|-----------------------|
| 0 | Agent | 0.036556 | 0.036556 | no |
| 1 | SGD | 0.033533 | 0.489764 | yes |
| 2 | Adadelta | 0.028980 | 0.157204 | yes |
| 3 | Adam | 0.133529 | 2.228178 | yes |
| 4 | SGD-Sch | 0.071715 | 0.071715 | no |
| 5 | Adadelta-Sch | 0.591062 | 0.591062 | no |
| 6 | Adam-Sch | 0.024535 | 0.024535 | no |

Table 9: Scheduler - OneCycleLR — Agent LR = 1e-2

|   | Method | Best Final Loss | Avg Final Loss | Divergent case exists |
|---|--------|-----------------|----------------|-----------------------|
| 0 | Agent | 0.509641 | 0.509641 | no |
| 1 | SGD | 0.031223 | 0.322218 | yes |
| 2 | Adadelta | 0.030701 | 0.155962 | yes |
| 3 | Adam | 0.115674 | 2.234299 | yes |
| 4 | SGD-Sch | 0.088461 | 0.088461 | no |
| 5 | Adadelta-Sch | 0.685073 | 0.685073 | no |
| 6 | Adam-Sch | 0.027269 | 0.027269 | no |

A.5    EXAMPLES OF DIVERGENT SCHEDULES ACROSS OPTIMIZER SCHEDULER CHOICES

We also note the no-free-lunch theorem by Wolpert , where no algorithm (here, a combination of optimizer-scheduler) can have a lower error overall, across all datasets and parameters. Hoewver, we are motivated by results that show that at the very least, non-divergent schedules are found by the agent. This overall stability is noteworthy and useful to study in a more theoretical setting, motivated by these intial experimental results.

Table 10: Scheduler - OneCycleLR — Agent LR = 5e-3 for CIFAR100

|   | Method | Best Final Loss | Avg Final Loss | Divergent case exists |
|---|--------|-----------------|----------------|-----------------------|
| 0 | Agent | 4.606702 | 4.606702 | no |
| 1 | SGD | 4.042578 | 4.788674 | yes |
| 2 | Adadelta | 3.161358 | 10.678298 | yes |
| 3 | Adam | 4.176720 | 4.678715 | yes |
| 4 | SGD-Sch | 3.436341 | 3.436341 | no |
| 5 | Adadelta-Sch | 4.553504 | 4.553504 | no |
| 6 | Adam-Sch | 4.869463 | 4.869463 | yes |

A.6 SOME LOSS CURVES AND GENERATED SCHEDULES

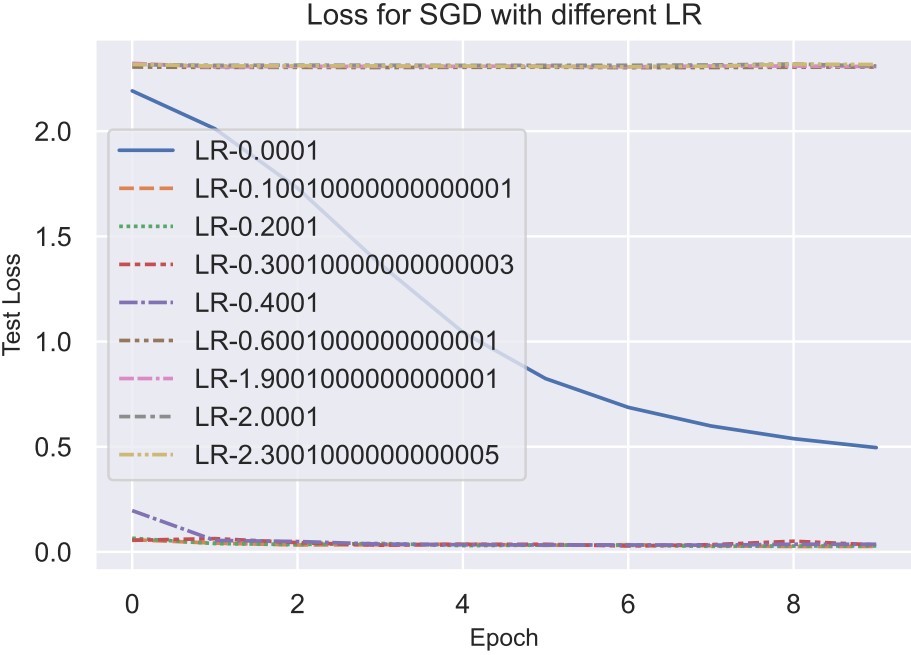

Figure 1: Training loss for LRs as part of grid search for SGD. Similar results were recorded for Adam, Adadelta and Adagrad but are not included here for brevity. The key takeaway is that grid search, and other HPO algorithms will discover both converging and diverging loss trajectories using initial, fixed LRs; i.e. some initial LRs are bound to diverge, regardless of the schedule (see figure 2 below). A learned schedule can result in stable loss curves, even from a difficult initial point

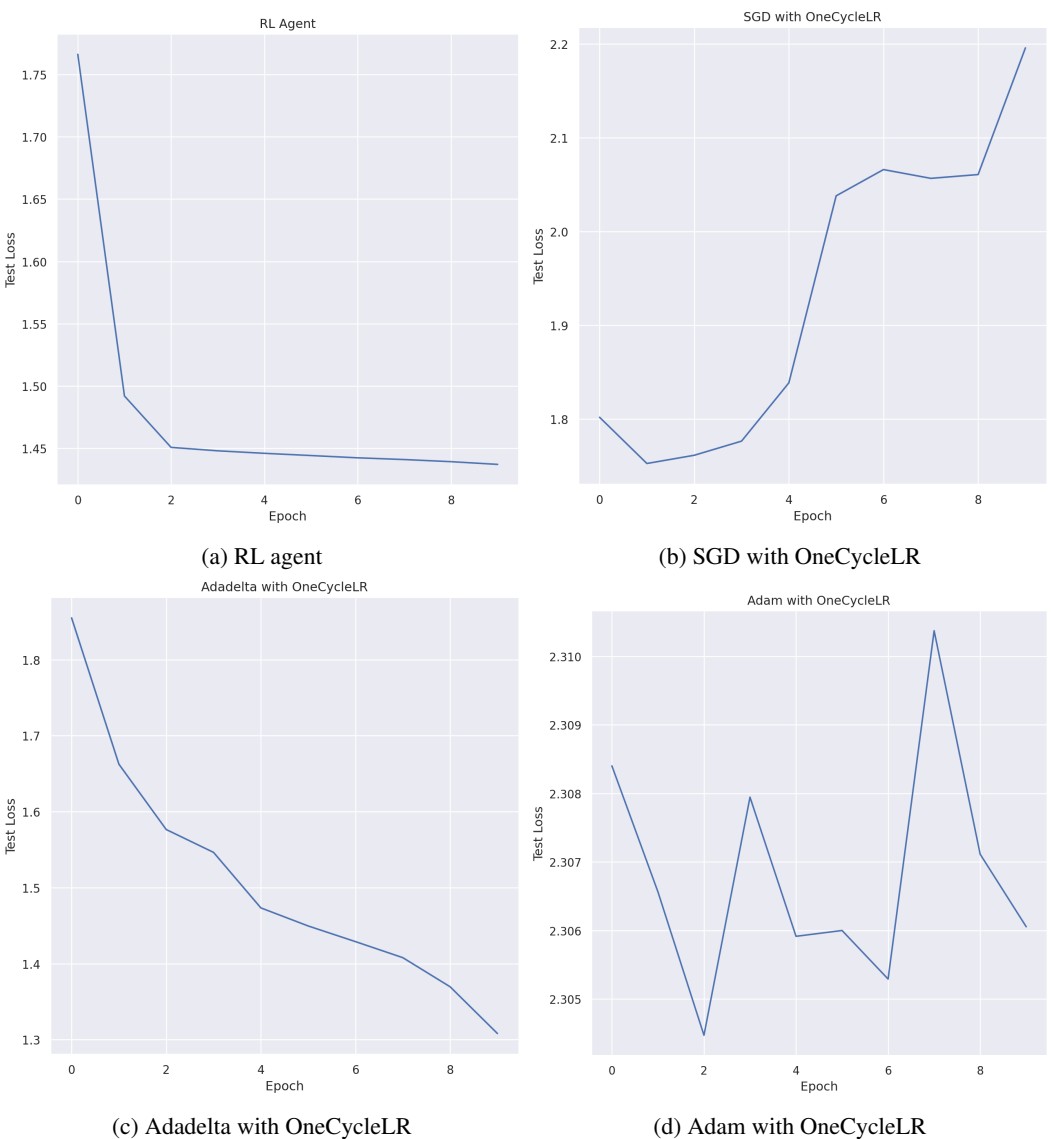

Figure 2: Validation losses on CIFAR-100 image classification dataset for the proposed RL agent along with popular optimizers with different learning rates. We see that the agent has a typical looking loss hostory, but may not beat the best, highly tuned optimizer scheduler combination. Others (like b) and d) can show erratic, divergent behavior, whereas the agent performs consistently, and is more stable on average across multiple runs.

A.7   COMPARISON WITH FULL LINE-SEARCH METHODS.

Recently Vaswani et al. (2019) (called `SLS`) developed an optimizer for SGD that automatically sets the step size of SGD by performing a line search at each iteration of training. Given that both `SLS` and the RL agent have access to similar information at inference time, we wanted to evaluate the performance of the agent with the `SLS` optimizer and compare their respective learning rate schedules. In Figure 6 we compare `SLS` with our learned optimizer and observe that the loss curve upper bounds that of `SLS`. However, we see no common discernible pattern in the step sizes proposed by the two algorithms. We made the following observations from this experiment **a)** step sizes set by the two optimizers were very different, despite having similar performances could suggest that there can be several "good" learning schedules for a given task; **b)** it is not clear as to how to compare

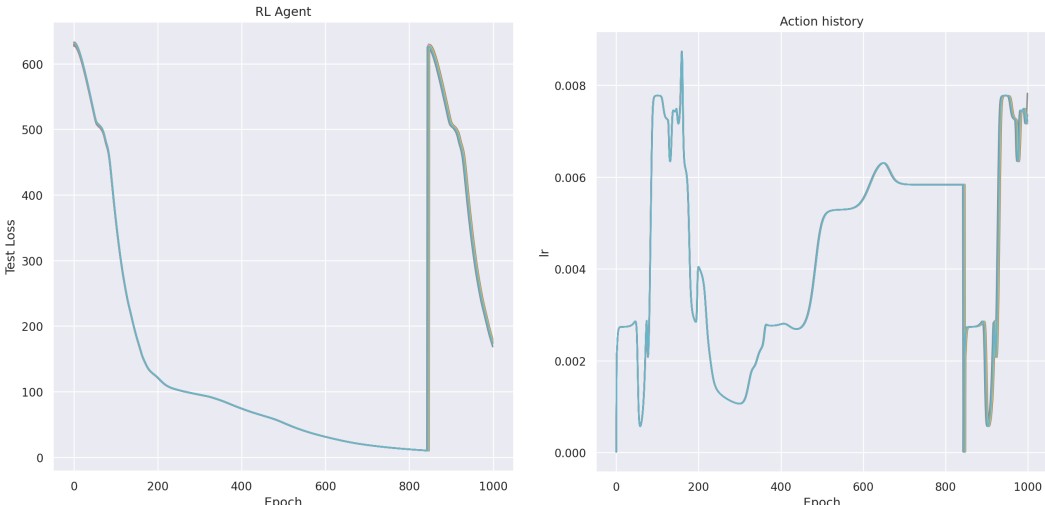

Figure 3: Validation loss and action histories of the proposed agent after $10K$ training iterations on MNIST image classification task.

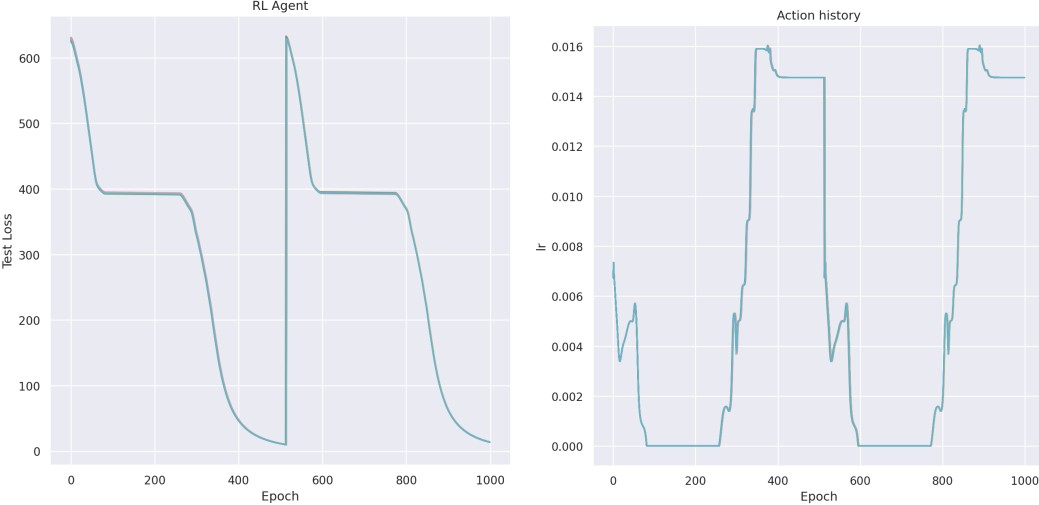

Figure 4: Validation loss and action histories of the proposed agent after $20K$ training iterations on MNIST image classification task.

two learning schedules rigorously for stochastic optimizers. We delegate these questions for future investigation.

### A.8 ABLATION EXPERIMENTS

**Impact of simplifying reward function.** We first simplify the reward function and remove the dependence on epochs (i.e., we set $\lambda = 0$):

$$r = \gamma(\texttt{best\_loss} - \texttt{val\_loss}) + \lambda(epoch_{\max} - epoch), \tag{3}$$

We also disregard best loss and set $\gamma = 1$ as with our original experiments.

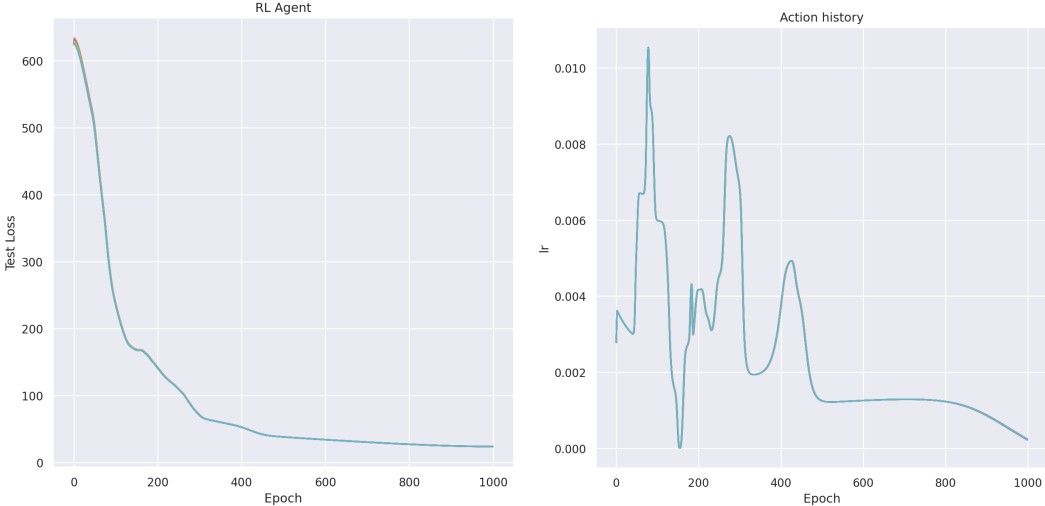

Figure 5: Validation loss and action histories of the proposed agent after $50K$ training iterations on MNIST image classification task.

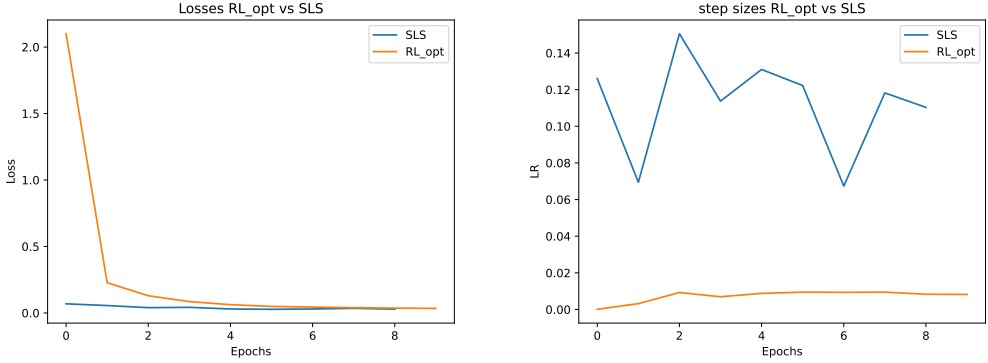

Figure 6: Our learned optimizer vs full line-search method of Vaswani et al. (2019) (`SLS`). (Left) Comparison of validation losses (right) comparison of learning rates predicted by the optimizers.

Comparing SGD performance overall by running a grid search over lr in [0,1] range, with average performance of learned agent runs trained over the same range of LRs, and with reward function being directly validation loss yields interesting results:

1. Average min test loss for SGD across all LRs is 0.36, vs 0.02 for the agent. This is consistent with our results for the full reward function; i.e., average performance for the agent is an order of magnitude better than the average performance for the optimizer chosen

2. Loss curves for SGD as expected can be highly lr dependent (Fig. 7)

3. Loss curves for the agent were more consistent, but different (Fig. 8)

4. Interesting that the action (lr) curves start low and go high when the agent reward was just loss, vs previous experiments with complex reward function (escape divergence + loss + monotonic etc) where loss started higher and went to lower bound (Fig. 9)

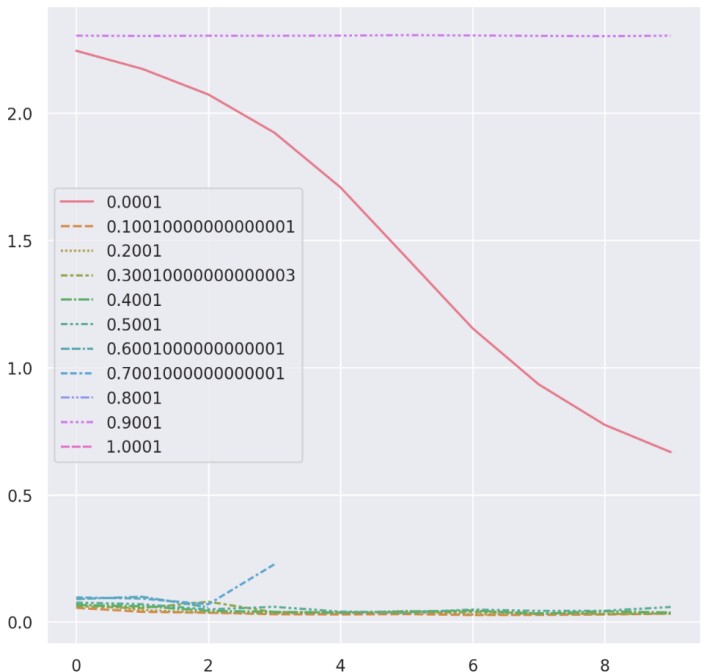

Figure 7: SGD grid search with a smaller range of LRs, compared to Fig. 1.

**Impact of changing learning rate for the agent.** We test three additional learning rates for the agent - 0.01, 0.05, 0.1, with all other factors remaining constant. Change in the LR for the agent (note, not the LR for actual model training, but for RL agent) impacts the final policy learned.

Action histories that A) start with a low LR and end higher seem to give better performance than the ones that B) start high and snap to the lower bound of LR. Higher learning rates for the RL agent end up in B, and performance gets worse as we increase the LR further.

## A.9 ACKNOWLEDGEMENTS

We would like to thank Randy DeFauw for his initial collaboration and brainstorming around this topic.

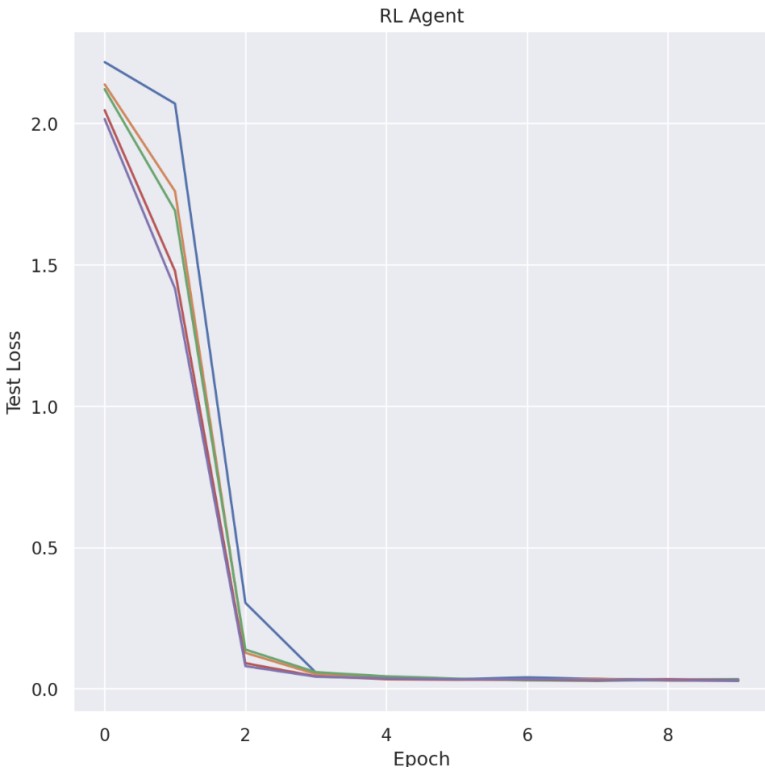

Figure 8: Agent loss histories (LR schedules) for the trained agent for 5 different seeds

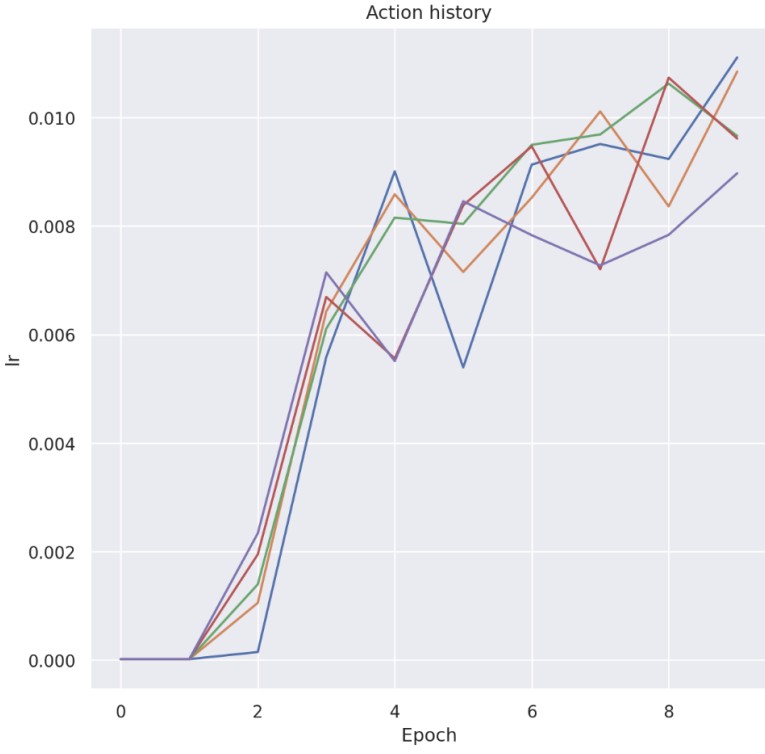

Figure 9: Agent action histories (LR schedules) for the trained agent for 5 different seeds

