# OpenReview forum: "Learned Learning Rate Schedules for Deep Neural Network Training Using Reinforcement Learning"
_ICLR.cc/2023/TinyPapers — Submitted to Tiny Papers @ ICLR 2023_

### Official Review · Reviewer_tYkE · 2023-03-30

**Confidence:** 4

**Summary Of Contributions:**

The paper proposes training a RL agent to decide the learning rate schedule of an optimizer. The agent receives a reward for quickly lowering the validation loss while being stable. They compare the proposed method against prior fixed schedules for learning rates such as one cycle LR.

**Rating:**

Needs Clarification (NC): a submission which does not meet the reviewing criteria and needs clarification for its described problem or solution

**Strengths And Weaknesses:**

Strengths
- The problem is of relevance to the community. RL does present a potential way to tune the learning rate of the optimizer for better performance.
- The paper shows promise as an initial step for better learning rate schedulers since they perform better than OneCycleLR on the MNIST classification task.

Weaknesses
- See suggested changes.

**Suggested Changes:**

## Major changes
- Table 1 shows that SGD without any learning rate schedule does better than the proposed method and OneCycleLR. So it is difficult to justify using RL for this task. One way you can justify it is if the grid search for LRs required more compute than the RL training. However the authors have not mentioned this. This would strengthen their claims.
- The authors should compare the computational cost of the proposed methods. RL can be sample inefficient, and in this setting one episode is one training run. So the time to train the RL agent can be so large than naive grid search is a much better solution.
- In a similar vein, the authors have not mentioned much about the training details for the RL agent (for eg, final number of training steps for the results, reward obtained at the end of training). I would like to see these details in the paper, either in the main text or the appendix.
- The authors have said that the proposed method is sensitive to the RL agent hyperparameters, but have not included the final hyperparameters used for training. This can make it difficult to reproduce their results.
- Only one seed/run has been presented in the paper. It would be better to run multiple seeds (for eg. 5 seeds) and present the result with error bars.
- For the comparison with SRS, it is difficult to discern the difference in final validation loss between the RL agent and SRS from the plot. Presenting it in a table would aid the presentation.

## Minor Concerns
- The citation style in the text seems to be incorrect. The cited work should appear in parenthesis, for example. (Metz et al 2019), unless it is used as a noun in the sentence.

---

> ### Author Response · Authors · 2023-05-31
> **Response to suggested changes**
>
> ### Summary
>
> Thank you for reviewing the paper and for providing your valuable suggestions. As seen in the detailed comments below, we have reported the results of several more experiments, and addressed suggested changes. As a result of addressing your comments and running additional experiments, we are much more confident in our hypothesis that RL agents as schedulers can be an equally important method for tuning neural networks compared to hyperparameter optimization.
>
> ---------------
>
> ### Response to Major changes
>
> Original suggestions are _italicized_, with our responses below each suggestion:
>
> * _Table 1 shows that SGD without any learning rate schedule does better than the proposed method and OneCycleLR. So it is difficult to justify using RL for this task. One way you can justify it is if the grid search for LRs required more compute than the RL training. However the authors have not mentioned this. This would strengthen their claims._
>     * Thank you for your comments on comparing with grid search and computational cost below. We have redone our experiments  by providing equal computational budget for RL and grid search for four different optimizers. We show that with equal computational budget (10K total optimizer steps), RL has the potential to find unique schedules that perform well on average, and avoid divergence. We see that while some runs may outperform the agent, the optimizer with fixed LR may underperform on average when compared to average runs of the trained RL agent. These new results are added to the main text and appendix.
>
> * _The authors should compare the computational cost of the proposed methods. RL can be sample inefficient, and in this setting one episode is one training run. So the time to train the RL agent can be so large than naive grid search is a much better solution._
>     * To make the comparison fair, we have used the exact same computational budget in order to prove that RL, and the goal of finding good schedules rather than a set of fixed hyperparameters can potentially be a better use of resources.
> * In a similar vein, the authors have not mentioned much about the training details for the RL agent (for eg, final number of training steps for the results, reward obtained at the end of training). I would like to see these details in the paper, either in the main text or the appendix.
>     * We have added more details on the training steps, agent used, and reward as part of the paper to clarify and highlight the above points
>
> * _The authors have said that the proposed method is sensitive to the RL agent hyperparameters, but have not included the final hyperparameters used for training. This can make it difficult to reproduce their results._
>     * We now include the final hyperparameters used and plan to release the code for the experiments and the raw results for experiments with two datasets, four optimizers, 2 schedulers, 10K steps and the trained agent weights
>
> * _Only one seed/run has been presented in the paper. It would be better to run multiple seeds (for eg. 5 seeds) and present the result with error bars._
>     * We include results of training with multiple seeds (i.e. every episode resets the seed, at most after max epochs of 100, and at least after one epoch of training). For the agent, we report average results after 5 random runs which in the case, for the reported resolution, matches the result for the best run as the agent learns a fixed policy. This is interesting as the agent may provide on-average more stable convergence, but may not result in the best validation loss, which we see via grid search or other optimizer-scheduler combinations. We observe that in these cases, where the best run is not of the agents’, the corresponding on-average performance for the other optimizer/scheduler combinations is poor. We have highlighted these results as suggested by your questions in the main paper and included more experiments in the appendix.
>
> * _For the comparison with SRS, it is difficult to discern the difference in final validation loss between the RL agent and SRS from the plot. Presenting it in a table would aid the presentation._
>     * We have moved comparisons with SLS to the appendix, as it may be interesting to some researchers who are looking to compare with exact line search methods for calculating optimal step size. We have instead brought to the main stage comparisons with commonly used schedulers like constantLR and OneCycleLR. We continue to run more experiments on ablating parts of the agent used, running for more iterations, and testing with other datasets, optimizers and schedulers which we plan to document in a future version of this paper.
> --------
> ### Response to Minor Concerns
> * _The citation style in the text seems to be incorrect. The cited work should appear in parenthesis, for example. (Metz et al 2019), unless it is used as a noun in the sentence._
>     * Thank you for pointing this out - we have now fixed the citation style

---

### Official Review · Reviewer_jakf · 2023-04-02

**Confidence:** 4

**Summary Of Contributions:**

The paper proposed using RL (specifically PPO algorithm) to generate learned learning rate schedules for SGD. It demonstrated faster convergence compared with fixed learning rate scheduler on image classification tasks.

**Rating:**

Great Start (GS): a submission which meets some of the reviewing criteria but has room for improvement

**Strengths And Weaknesses:**

## Strength:
1. Interesting application of RL on an important problem of learning rate scheduler.
2. Demonstrated faster convergence on MNIST and CIFAR-100 datasets.
## Weaknesses:
1. Baseline of SLS is a bit weak. It would be interesting to see compare it with some other learning rate schedulers. Only the final loss is not intuitive.
2. The number of epochs is relatively small for all the experiments.
3. The tasks are not performed on larger datasets. Also, it's unclear how different batch sizes will affect the conclusion/observations.
4. It's unclear whether the problem has an MDP structure since the state and action spaces are not defined. It will be better to demonstrate the parameters and network architectures chosen for PPO as well.



**Suggested Changes:**

## Suggested changes
1. Make some ablation studies on the reward, and state space formulation.
2. It is important to show the formulation of MDP on this problem and show how your choices generalize to different tasks and would be reproducible in different settings (e.g. batch sizes).
3. Clearly state the gain on wall-clock time for training to a certain accuracy (time-to-accuracy), and final test accuracy,  and show more epochs on larger datasets. These metrics will make the contribution much clearer.
4. Generate some other learning rate schedules with different algorithms other than PPO (maybe TRPO or A3C), what kind of results do you see?
5. On the reward formulation, it's not necessary to use two weighting parameters. It would be interesting to show some results by changing these parameters.

---

> ### Author Response · Authors · 2023-05-31
> **Response to suggestions**
>
> ### Response to suggested changes
>
> 1. _Make some ablation studies on the reward, and state space formulation_.
>     1. We have added ablation results for simplifying the reward signal, and for changing the learning rate of the agent. While we are working on completing more results for ablation, we have added results that compare RL and HPO (grid search) with equal budgets (i.e. number of optimizer steps), compared with two new schedulers, and ran the experiments for 10x number of epochs. We have reported results of our large number of experiments in the main paper and the appendix, and continue to work on further experiments, datasets, agent architectures and reward signals
>
> 2. _It is important to show the formulation of MDP on this problem and show how your choices generalize to different tasks and would be reproducible in different settings (e.g. batch sizes)._
>     1. We include the formal POMDP formulation in Appendix A.1 and added more experiments and ablation results. We include more detail in our experiments to make sure that these results are reproducible as suggested.
>
> 3. _Clearly state the gain on wall-clock time for training to a certain accuracy (time-to-accuracy), and final test accuracy, and show more epochs on larger datasets. These metrics will make the contribution much clearer._
>     1. As a result of your suggestions and the other reviewers’, we train to 10x more epochs (100 instead of 10), provide a fair comparison by allowing a maximum computational budget for the total optimizer steps allowed by the agent as well as for full exhaustive grid search (10000). We observe that the wall clock time for the trained agent when used for purely generating schedules for an optimizer has no clear advantage, however, the agent does generate stable trajectories compared to the sometimes diverging results when combining common optimizers and schedulers as discussed in the results.
>
> 4. _Generate some other learning rate schedules with different algorithms other than PPO (maybe TRPO or A3C), what kind of results do you see?_
>     1. We are currently in the process of repeating all these results with TRPO and A3C, but owing to the fact that these experiments, even for small model networks are expensive to run, we are still not ready to publish these results. These experiments are still ongoing. Currently  we include experiments for 2 datasets, 2 schedulers and 4 optimizers with full grid search, as well as agent steps to match the grid search results. We look forward to testing further and publishing the same at a future venue
>
> 5. _On the reward formulation, it's not necessary to use two weighting parameters. It would be interesting to show some results by changing these parameters._
>     1. We have changed our results to use only one weight parameter (i.e. we set Lambda to 1) in all experiments. We are still working on quantitatively assessing the impact of the weight parameter along with ablating reward signal. We do include a section on ablation in the appendix where we set Gamma to zero, and directly minimize validation loss. We see results that are similar to the full reward function and highlight some differences in the appendix

---

> ### Author Response · Authors · 2023-05-31
> **Response summary and response to weaknesses**
>
> ### Summary
>
> We believe we have addressed all your valuable points in the weaknesses and suggestions. We add comment-specific responses below and point you to the latest paper version above for clarification. Original comments are _italicized_ below and one or more response points are enumerated below the original comments:
>
> ---------------
>
> ### Response to weaknesses
> 1. _Baseline of SLS is a bit weak. It would be interesting to see compare it with some other learning rate schedulers. Only the final loss is not intuitive._
>     1. As mentioned in the paper, we have now added experiments with 4 baseline optimizers and two schedulers - ConstantLR and OneCycleLR as implemented in pytorch’s torch.optim class - https://pytorch.org/docs/stable/optim.html
>     2. We include the SLS comparisons as additional results in the appendix, since SLS provides the theoretical optimum LR through line search and this may be interesting to some researchers
>
> 2. _The number of epochs is relatively small for all the experiments._
>     1. We agree that our initial number of epochs of 10 was very low - we have increased the number of epochs of each run from 10 to 100 (10x) and would like to note that each comparison involves doing 1) full grid search for each of the base optimizers,2)  running the base optimizers with the chosen schedulers, 3) training the agent and 4) evaluating the trained agent 5 times to report average results.
>
> 3. _The tasks are not performed on larger datasets. Also, it's unclear how different batch sizes will affect the conclusion/observations._
>     1. We include results for larger datasets and also ablating other important factors such as the reward signal, and learning rate for the agent. We would like to kindly note that results with these smaller datasets are themselves very costly, as explained in 2a. We are working on even larger datasets, varying batch sizes for both the RL workers as well as the base network, and including more task types and models (NLP, Large language model tasks) as a next step.
>
> 4. _It's unclear whether the problem has an MDP structure since the state and action spaces are not defined. It will be better to demonstrate the parameters and network architectures chosen for PPO as well._
>     1. We have rewritten the beginning of section 2 describing the state, observation and action spaces. Since we only observe the validation loss, remaining epochs and gradient norms, the environment is only partially observable in our set up. The action space (box) outputs a float LR value that is fed into the optimizer at each step. As seen in proofs of convergence for SGD and other optimizers, the SGD step can be independently modeled as a POMDP.
>
> ---------------

---

### Author Response · Authors · 2023-05-31
**Archival opt in**

Please note that we wish to opt-in for archival

---

### Meta-Review · Area_Chair_9UJF · 2023-04-06

**Recommendation:** Invite to revise
**Confidence:** 5

**Metareview:**

The author proposes a RL-based scheduler to determine the optimal learning rate. However, this work is not novel and not solid. Also, it misses theoretical proof. Also, the presentation is poor.

**Summary:**

The work proposes a learning rate scheduler for RL.

**Comments And Feedback To The Authors:**

Please refer to the meta review.

**Reason For Not Giving A Higher Recommendation:**

This work is not novel and not solid.

**Reason For Not Giving A Lower Recommendation:**

N/A

---

> ### Author Response · Authors · 2023-05-31
> **Response to meta review**
>
> We thank you and the reviewers for your valuable comments to make our contributions stronger. Based on the suggestions, we have added several more experiments involving more optimizer-scheduler combinations, training for larger number of epochs, comparing with grid search with an equal computational budget, formulating the problem as a POMDP and responding to each of the reviewers comments with changes in the main text and the appendix which also includes running additional experiments. We appreciate the feedback and believe that with the several corrections and experimental additions, our contributions are clearer and stronger. Specifically, when comparing hyperparameter optimization algorithms (grid search being an exhaustive search) with RL that has access to the same overall computational budget, RL agents as schedulers can show promising results by producing dynamic, problem-specific Learning rate schedules that are performant and more stable in practice.

---

### Decision · Program_Chairs · 2023-04-08

Revision accepted; invite to archive